# Antifungal Activity of Selected Naphthoquinones and Their Synergistic Combination with Amphotericin B Against *Cryptococcus neoformans* H99

**DOI:** 10.3390/antibiotics14060602

**Published:** 2025-06-13

**Authors:** Naira Sulany Oliveira de Sousa, Juan Diego Ribeiro de Almeida, Linnek Silva da Rocha, Izabela de Mesquita Bárcia Moreira, Flávia da Silva Fernandes, Ani Beatriz Jackisch Matsuura, Kátia Santana Cruz, Emersom Silva Lima, Érica Simplício de Souza, Hagen Frickmann, João Vicente Braga de Souza

**Affiliations:** 1Program in Biodiversity and Biotechnology of the Bionorte Network (PPG-BIONORTE), Amazonas State University (UEA), Manaus 69065-001, AM, Brazil; nsods.dbb21@uea.edu.br; 2Mycology Laboratory, National Institute for Amazonian Research (INPA), Manaus 69067-375, AM, Brazil; jdra.dbm23@uea.edu.br (J.D.R.d.A.); lsr.mbt23@uea.edu.br (L.S.d.R.); idmbm.dbb23@uea.edu.br (I.d.M.B.M.); flaviafernandes19@gmail.com (F.d.S.F.); 3Leônidas & Maria Deane Research Center, Oswaldo Cruz Foundation (FIOCRUZ), Manaus 69057-070, AM, Brazil; ani.matsuura@fiocruz.br; 4Mycology Laboratory, Dr. Heitor Vieira Dourado Tropical Medicine Foundation (FMT-HVD), Manaus 69040-000, AM, Brazil; katia.cruz@fmt.am.gov.br; 5Faculty of Pharmaceutical Sciences, Federal University of Amazonas (UFAM), Manaus 69067-005, AM, Brazil; eslima@ufam.edu.br; 6Higher School of Technology, Amazonas State University (UEA), Manaus 69050-020, AM, Brazil; esdsouza@uea.edu.br; 7Department of Medical Microbiology, Virology and Hygiene, University Medicine Rostock, 18057 Rostock, Germany; frickmann@bnitm.de; 8Department of Microbiology and Hospital Hygiene, Bundeswehr Hospital Hamburg, 22049 Hamburg, Germany

**Keywords:** *Cryptococcus*, naphthoquinones, amphotericin B, antifungal synergy

## Abstract

**Background/Objectives:** Cryptococcosis, caused by *Cryptococcus neoformans* and *Cryptococcus gattii* species complexes, remains a significant health concern, particularly among immunocompromised patients. The emergence of antifungal resistance and toxicity of conventional treatment underscore the urgent need for novel therapeutic approaches. Combination therapies represent a promising strategy to enhance efficacy and overcome resistance. This study investigated the antifungal activity of five naphthoquinones against nine isolates of *Cryptococcus* spp. and assessed their synergistic effects with amphotericin B (AmB). **Methods**: In this study, five selected naphthoquinones were evaluated for their antifungal activity against *Cryptococcus* spp. isolates using broth microdilution assays to determine minimum inhibitory concentrations (MICs), according to CLSI guidelines. The potential synergistic effect with AmB was assessed using checkerboard assays, with synergy interpreted based on the fractional inhibitory concentration index (FICI). Cytotoxicity was evaluated in MRC-5 human lung fibroblast cells using the MTT assay. **Results**: Among the compounds tested, 2-methoxynaphthalene-1,4-dione (2-MNQ) demonstrated antifungal activity, with MIC values ranging from 3.12 to 12.5 µg/mL. Checkerboard assays revealed a synergistic interaction between 2-MNQ and AmB, with a fractional inhibitory concentration index (FICI) of 0.27. The combination reduced the MIC of AmB by 4.17-fold. These findings highlight the potential of synthetic naphthoquinones, particularly 2-MNQ, as effective antifungal agents with synergistic properties when combined with AmB. The observed synergy suggests complementary mechanisms, including increased fungal membrane permeability and oxidative stress induction. **Conclusions**: This study highlights the potential of 2-MNQ and 2,3-DBNQ as antifungal candidates against *Cryptococcus* spp., with emphasis on the synergistic interaction observed between 2-MNQ and amphotericin B. The findings reinforce the importance of structural modifications in naphthoquinones to enhance antifungal activity and support the need for further preclinical studies investigating combination therapies aimed at improving treatment efficacy in patients with cryptococcosis.

## 1. Introduction

Cryptococcosis, caused by *Cryptococcus* spp., is a life-threatening fungal infection that predominantly affects immunocompromised individuals, such as those suffering from HIV/AIDS (human immunodeficiency virus/acquired immunodeficiency syndrome) [1,2]. It has been associated with more than 181,000 deaths annually, and the highest incidence has been reported for sub-Saharan Africa and parts of Latin America. The opportunistic infection shows a tropism towards the central nervous system, often resulting in meningoencephalitis, a condition with high mortality rates in the absence of early and adequate medical treatment [3,4]. Despite advances in antifungal therapy, current treatment regiments including amphotericin B remain limited due to their toxicity, high costs, and restricted availability in low-resource settings [5]. Moreover, the growing prevalence of antifungal resistance implies additional challenges [2], emphasizing an urgent need for innovative approaches in order to address these limitations.

Research into alternative therapeutic approaches has suggested a potential of naphthoquinones, a class of compounds known for their antimicrobial, antitumor, and antiparasitic properties [6,7,8,9,10,11]. These molecules have demonstrated potent antifungal activity against *Cryptococcus* spp. Their described molecular mechanisms included membrane disruption and enzyme inhibition [12,13]. Systematic reviews have summarized previous findings on their structural diversity and bioactivity, underscoring a likely future relevance for antifungal drug development [13,14]. However, despite these encouraging results, the progression from in vitro studies to clinical applications remains hindered by a lack of comprehensive investigations into their pharmacological properties and toxicity.

While the synergy of established antifungal combinations, such as amphotericin B and flucytosine, has significantly improved treatment outcomes [15], the exploration of similar synergistic effects with compounds like naphthoquinones has so far been notably scarce [14,16]. This gap is particularly striking considering the potential of synergistic therapies for enhancing therapeutic efficacy, reducing adverse effects, and counteracting resistance [17,18,19,20]. Addressing this lack of available research data could pave the way towards more effective and accessible treatments for cryptococcosis, especially in regions affected by resource constraints and high disease prevalence.

This study evaluated the antifungal activity of selected naphthoquinones and their interactions with amphotericin B when applied as therapeutically active agents against *Cryptococcus* spp. The study objectives include determining minimum inhibitory concentrations (MICs), analyzing synergistic potential using checkerboard assays, and assessing cytotoxicity in human fibroblast cells. By identifying therapeutic combinations which achieve relevant fungal inhibition while maintaining low toxicity, this research aims at contributing to the development of new therapeutic strategies against *Cryptococcus* spp.

## 2. Results

### 2.1. Antifungal Activity

Five naphthoquinones were evaluated for their antifungal activity against nine selected isolates of the *Cryptococcus neoformans*/*gattii* complex. The naphthoquinones showing the lowest MIC values were 2,3-DBNQ (MIC = 0.19 µg/mL) and 2-MNQ (MIC = 3.12 to 12.5 µg/mL). The remaining naphthoquinones tested (lapachol, 2-ClFNQ, and atovaquone) inhibited *Cryptococcus* strains with MIC values ≥ 100 µg/mL only (Table 1).

### 2.2. Interaction Studies

#### 2.2.1. Naphthoquinones That Enhance the Antifungal Activity of Amphotericin B

The growth kinetics of the selected *Cryptococcus neoformans* H99 strain were evaluated (Figure 1). The growth curve indicates that the lag phase lasted approximately 24 h, while stabilization was observed after 72 h of cultivation. To determine the subinhibitory concentration of amphotericin B for *C. neoformans* H99, a dose–response assay was conducted. Complete growth inhibition was observed at concentrations ≥ 0.5 µg/mL. Amphotericin B at 0.03 µg/mL resulted in a 20% reduction in fungal growth. The concentration 0.03 µg/mL was selected as the minimal subinhibitory concentration for use in the screening assays together with naphthoquinones that were assessed for a potential enhancement of the antifungal activity of amphotericin B.

A screening was conducted to identify which naphthoquinones enhance the inhibitory effect of amphotericin B against *Cryptococcus neoformans* H99. The combinations of amphotericin B with atovaquone, 2-ClFNQ, and 2,3-DBNQ resulted in fungal growth comparable to the untreated control (Figure 2). The combination with lapachol showed an inhibition of 42.57%, corresponding to 57.43% fungal survival. Among the compounds tested, 2-MNQ showed the most effective and synergistic profile. The combination of 2-MNQ (1 µg/mL) with amphotericin B (0.03 µg/mL) showed high efficacy during the first 48 h of incubation, with an inhibition of 94.72%, compared to 63.84% observed for 2-MNQ alone. After 72 h, both conditions showed a reduction in inhibition, with values of 82.14% for the combination and 32.20% for 2-MNQ alone. Erythromycin (18 µg/mL), used as a control for synergy, in combination with amphotericin B (0.03 µg/mL), inhibited 99.13% and 95.38% of fungal growth after 48 and 72 h, respectively (Figure 2).

#### 2.2.2. Interaction Profiles from the Checkerboard Assay

To confirm the results of the preliminary screening, 2-MNQ and amphotericin B were tested at various combined concentrations against the *C. neoformans* H99 strain using the checkerboard method. The combination of 0.12 µg/mL of amphotericin B with 0.12 µg/mL of 2-MNQ was the lowest concentration that inhibited fungal growth, reducing the MIC of amphotericin B from 0.5 µg/mL to 0.12 µg/mL, a 4.17-fold decrease. The FICI was calculated at 0.27, highlighting a synergistic interaction between the compounds (Figure 3).

### 2.3. Toxicity Assays

MRC-5 fibroblasts, derived from human lung tissue, were treated with different concentrations of selected naphthoquinones (100, 50, 25, 12.5, 6.25, and 3.125 μM/mL) for a period of 24 h. The IC50 values and corresponding masses for the naphthoquinones were as follows: 2-methoxynaphthalene-1,4-dione (2-MNQ)—11.94 μM (2.25 μg/mL), 2,3-dibromonaphthalene-1,4-dione (2,3-DBNQ)—15.44 μM (4.88 μg/mL), and 2-chloro-3-(2-fluoroanilino)naphthalene-1,4-dione (2-ClFNQ)—29.29 μM (7.09 μg/mL). Lapachol demonstrated low cytotoxicity in the MRC-5 fibroblast model. Even at the highest concentration tested (100 μM, corresponding to 30.17 μg/mL), the compound maintained 65% cell viability.

## 3. Discussion

This study provides in vitro evidence of the antifungal activity of the selected naphthoquinones against the *Cryptococcus neoformans* complex as well as the *Cryptococcus gattii* complex and of the particularly pronounced activity of the synthetic compounds 2,3-DBNQ and 2-MNQ. These compounds exhibited the lowest MIC values of 0.19 µg/mL and 3.12 µg/mL. This highlights their increased antifungal potency compared to natural derivatives such as lapachol, which showed an MIC of 1.000 µg/mL. Furthermore, the experiments identified 2-MNQ as a potential key candidate for combination, demonstrating a substantial synergistic interaction with amphotericin B. The checkerboard assay confirmed this synergy with a FICI of 0.27. These findings indicate that structural modifications in synthetic naphthoquinones play a crucial role in enhancing antifungal efficacy and open new perspectives on the development and evaluation of effective combination therapies for the treatment of cryptococcosis. Together, these results underline a potential of 2-MNQ as a novel therapeutic agent with likely benefits if used alone or in combination.

After 48 h, the combination of 2-MNQ (1 µg/mL) with AmB (0.03 µg/mL) achieved a fungal growth inhibition of 94.72%, demonstrating a marked improvement compared to the 63.84% observed for 2-MNQ alone. This synergistic effect underscores the potential of combining synthetic naphthoquinones like 2-MNQ with established antifungal agents to enhance therapeutic efficacy. However, a decline in inhibitory activity was observed over time. After 72 h, the combination maintained an inhibition of 82.14%, while 2-MNQ alone showed a significant reduction to 32.20%. This decrease suggests that the antifungal activity of 2-MNQ may diminish with prolonged exposure, possibly due to metabolic degradation, fungal cell adaptation, or reduced compound stability under experimental conditions. Although the chemical stability of 2-MNQ was not directly assessed in this study, this limitation is acknowledged and discussed. Moreover, considering that AmB alone at the tested concentration (~0.03 µg/mL) resulted in only ~20% inhibition at both time points, it is unlikely that the sustained antifungal activity observed in the combination is solely attributable to AmB, further supporting a functional synergistic interaction [15,20].

The observed antifungal activity of naphthoquinones like 2,3-DBNQ and 2-MNQ confirms the importance of structural modifications in order to optimize their effects. Halogen atoms, such as bromine in 2,3-DBNQ, and methoxy groups in 2-MNQ enhance the lipophilicity and reactivity of these compounds and thus facilitate interactions with fungal cell membranes and mitochondria [21,22]. Previous studies confirmed these assumptions and showed that halogenation increases antifungal potency by improving compound permeability and target interaction [23,24]. In line with this, synthetic naphthoquinones demonstrated increased activity compared to natural derivatives like lapachol, which had limited efficacy (MIC = 1.000 µg/mL) in the here-presented study. Similar observations were made by Tran et al. [25], who also proved that structural modifications enhance the antifungal profile of quinone derivatives. Taken together, these findings support the chemical tailoring of naphthoquinones as an effective approach for improving antifungal activity against *Cryptococcus* spp.

The observed synergy between 2-MNQ and amphotericin B represents a contribution to the currently available information on therapeutic combination options. The checkerboard assay demonstrated a 4.17-fold reduction in the MIC of amphotericin B when combined with 2-MNQ, which was confirmed by a FICI of 0.27. Although the heatmap derived from the checkerboard assay provides an overview of the antifungal response across different concentrations, it may not always inform on the true nature of drug interactions. In pharmacological terms, an additive effect represents a baseline scenario where the combined action of two agents equals the sum of their individual effects. However, to objectively evaluate synergy, quantitative metrics such as the fractional inhibitory concentration index (FICI) are essential [26]. According to the criteria established by Odds [27], FICI values ≤ 0.5 indicate synergistic interactions. In our study, the combination of 2-MNQ and amphotericin B against *Cryptococcus neoformans* H99 yielded a FICI of 0.27, speaking in favor of a synergistic effect. This finding highlights the potential of 2-MNQ as an adjuvant compound capable of enhancing the antifungal efficacy of amphotericin B, potentially allowing for lower therapeutic doses and associated reduced toxicity.

This synergistic effect is likely driven by complementary mechanisms: amphotericin B disrupts fungal cell membranes through ergosterol binding [28], while 2-MNQ intensifies this effect by increasing membrane permeability and inducing oxidative stress [12]. Such dual-action strategies have proven effective in other antifungal therapy studies [20]. For example, the use of atorvastatin (ATO) combined with fluconazole (FLC) resulted in increased reactive oxygen species production and reduced fungal burden in the lungs and brain of infected mice [29]. Notably, 2-MNQ showed similar synergistic effects to erythromycin, a compound which has previously been shown to enhance amphotericin B activity [30]. The findings emphasize the potential of 2-MNQ as a synergistic agent, which might help to optimize antifungal regimens for cryptococcosis treatment in the future. On the other hand, three naphthoquinones, 2,3-DBNQ, 2-ClFNQ, and atovaquone, exhibited antagonistic interactions with amphotericin B, as evidenced by fungal growth comparable to the untreated control. These results suggest that certain chemical structures and redox properties of naphthoquinones may interfere with the mechanism of action of amphotericin B, compromising its efficacy [22]. This underscores the need to investigate the specific molecular interactions of these derivatives in future studies.

To estimate the therapeutic potential of 2-MNQ, the selectivity index (SI) was calculated as the ratio between its IC_50_ in MRC-5 fibroblast cells (2.25 µg/mL) and its MIC against *Cryptococcus neoformans* (ranging from 3.12 to 12.5 µg/mL), yielding SI values from 0.18 to 0.72. While SI ≥ 10 is widely considered as the threshold for a favorable safety profile in early-stage drug screening [31,32], the concentration of 2-MNQ used in the synergy assays (1 µg/mL) was below the IC_50_ value, indicating that antifungal activity was achieved without exceeding in vitro cytotoxic limits. Despite the low SI values observed, the synergistic interaction with amphotericin B and the potential for structural optimization suggest further investigation of 2-MNQ as a lead compound in combined antifungal strategies.

The mechanism of action of naphthoquinones provides an explanation of their antifungal efficacy. These compounds generate reactive oxygen species (ROS) by redox cycling. This in turn leads to oxidative stress that destabilizes mitochondrial integrity and impairs cellular respiration [13]. This mechanism is well in line with studies highlighting the role of ROS in antifungal activity, such as the works by Shikov et al. [33] and Raghuveer et al. [34]. They demonstrated mitochondrial damage as a key target of oxidative stress in fungal cells. The here-proposed ability of 2-MNQ to complement amphotericin B by exacerbating membrane damage stresses its value for therapeutic combination. Comparative assessments by Navarro-Tovar et al. [22] suggest that such ROS-mediated mechanisms are crucial for overcoming fungal resistance and improving therapeutic outcomes. Future studies should focus on dissecting the molecular pathways in more detail, using in silico modeling, advanced biochemical assays, and investigations of in vitro mechanisms of action.

Despite the antifungal efficacy and observed in vitro synergy demonstrated in this study, it has several limitations that should be specifically addressed in future analyses. First, although likely mechanisms such as membrane disruption and oxidative stress have been suggested based on previous studies [12], our investigation did not include experiments aimed at directly confirming these pathways. Accordingly, the proposed mechanisms should be considered as hypotheses only, as they were not validated by applying molecular or biochemical assays. Therefore, no mechanistic conclusions can be drawn from the current dataset. To elucidate the molecular basis of the observed synergy, future studies should incorporate ROS quantification, membrane integrity assays (e.g., SYTO/PI staining), and ultrastructural analysis using electron microscopy (SEM or TEM) to visualize morphological features of fungal cell death (e.g., pyroptosis or necroptosis). These approaches would help to validate the proposed mode of action. Moreover, toxicity assessments were limited to a single human fibroblast cell line (MRC-5), which restricts the generalizability of the safety profile. Future investigations should include additional cell lines and primary human cells to provide a broader and more translational toxicological assessment, supporting the safe progression to in vivo models.

In addition, the chemical stability of 2-MNQ was not directly evaluated by applying the 72 h assays. This limitation may partly explain the reduction in antifungal activity observed over time and should be addressed in future studies. Despite these limitations, the demonstrated antifungal efficacy and in vitro synergy suggest that 2-MNQ may serve as a promising candidate for future combination therapy strategies in cryptococcosis, particularly in scenarios with limited access to other antifungal agents or in cases of rising drug resistance. Lastly, while the study included a selection of *Cryptococcus* spp. isolates, it did not capture the full genetic diversity of clinical and environmental isolates. Future research should aim at addressing these gaps by expanding the strain panel, conducting preclinical toxicity and efficacy studies, and exploring the broader applicability of naphthoquinones in combination therapies. Advances in understanding these aspects could significantly improve the translational potential of 2-MNQ and 2,3-DBNQ, providing potential options for novel antifungal therapies.

## 4. Materials and Methods

### 4.1. Microorganisms

The *Cryptococcus* spp. reference strains used in this study belong to the Fiocruz/Wieland Meyer Collection, a reference collection maintained by the Oswaldo Cruz Foundation (Fiocruz) in Brazil. This collection houses a diverse range of *Cryptococcus* spp. strains, including clinical, environmental, and veterinary isolates, widely used in research on fungal pathogenesis and antifungal susceptibility. These strains were kindly provided to the Mycology Laboratory of the National Institute of Amazonian Research (INPA) to support this study. The panel consisted of four *C. neoformans* strains—WM148 (serotype A, VNI), WM626 (serotype A, VNII), WM628 (serotype AD, VNIII), and WM629 (serotype D, VNIV)—and four *C. gattii* strains—WM179 (serotype B, VGI), WM178 (serotype B, VGII), WM161 (serotype B, VGIII), and WM779 (serotype C, VGIV). These molecular types are internationally recognized as the major lineages most frequently associated with human cryptococcosis, as demonstrated by molecular epidemiology studies and global surveillance data [35,36,37,38]. Their inclusion enabled a comparative assessment of antifungal susceptibility across genetically distinct and clinically relevant variants.

Prior to the start of each experiment, the yeasts were sub-cultured twice on Sabouraud agar (Himedia Laboratories, Kennett Square, PA, USA) to ensure both viability and culture purity [39]. *Cryptococcus neoformans* H99, a strain associated with clinical cryptococcosis and obtained from the American Type Culture Collection (ATCC), was used as a reference strain for susceptibility testing and synergism studies. All strains were stored in the INPA’s collection of medically relevant microorganisms in Sabouraud broth supplemented with glycerol at −20 °C.

### 4.2. Naphthoquinones and Drugs

All naphthoquinone derivatives used in this study were obtained from Sigma-Aldrich Corporation (St. Louis, MO, USA). The naphthoquinones used include 2-methoxynaphthalene-1,4-dione (2-MNQ), 2-chloro-3-(2-fluoroanilino)naphthalene-1,4-dione (2-ClFNQ), 2,3-dibromonaphthalene-1,4-dione (2,3-DBNQ), 4-hydroxy-3-(3-methylbut-2-enyl)naphthalene-1,2-dione (lapachol), and trans-2-[4-(4-chlorophenyl)cyclohexyl]-3-hydroxy-1,4-naphthoquinone (atovaquone). The chemical structures of these naphthoquinones are illustrated in Figure 4.

The naphthoquinones were dissolved in dimethyl sulfoxide (DMSO) to obtain 100-fold concentrated stock solutions and stored at −20 °C for use in the assays. For the assays, the stock solutions were further diluted in RPMI 1640 medium (Sigma-Aldrich Co., St. Louis, MO, USA) to achieve the desired concentrations, ranging from 1.56 to 800 μg/mL. The selection of these specific naphthoquinones was based on their previously described biological activities from the literature [12,40].

Amphotericin B (AMB) (Sigma-Aldrich Co., St. Louis, MO, USA) was dissolved in DMSO (Vetec, Rio de Janeiro, Brazil) and stored at −20 °C as a concentrated stock solution. The final concentrations used in the assays ranged from 16 to 0.03 μg/mL. Additionally, erythromycin (Sigma-Aldrich Co., St. Louis, MO, USA) was also stored as a concentrated stock solution at −20 °C. The selection of erythromycin (Ery) was based on its described synergistic effect with amphotericin B from the literature [30]. The final DMSO concentration in all dilutions was 1%.

### 4.3. Antifungal Activity Assay

The minimum inhibitory concentration (MIC) assays were performed in line with the broth microdilution method described by the CLSI (Clinical and Laboratory Standards Institute) in document M27-A4 [39]. Briefly, 100 μL of each naphthoquinone, diluted in RPMI 1640 broth buffered with 3-(N-morpholino) propanesulfonic acid (MOPS; 0.165 M, pH 7.0), was added to 96-well microplates, with final concentrations ranging from 800 to 1.56 μg/mL. Afterward, 100 μL of an inoculum containing 2.5 × 10^3^ yeast cells/mL was added to each well. The microdilution plates were incubated at 35 °C, and visual readings were performed after 72 h. The MIC was defined as the lowest concentration of naphthoquinone causing 100% inhibition of fungal growth. Amphotericin B was used as the control antifungal at concentrations ranging from 16 to 0.003 μg/mL. The negative control consisted of RPMI 1640 broth without naphthoquinones or other antifungal agents.

### 4.4. Screening and Interaction Studies

#### 4.4.1. Growth Curve Assessment with *Cryptococcus neoformans*

To assess the growth profile of *Cryptococcus neoformans* H99 as the chosen reference microorganism, the growth was monitored over time. The strain was cultured in RPMI medium buffered with MOPS [39] at an inoculum of 2.5 × 10^3^ CFU/mL. Incubation was performed at 35 °C with constant agitation at 100 rpm. Samples were collected after 0, 24, 48, 72, and 96 h of growth, and optical density was measured at 600 nm using a spectrophotometer (Eppendorf BioSpectrophotometer Kinetic, Hamburg, Germany), allowing the presentation of a growth curve to analyze the fungal proliferation rate [41]. Data analysis was performed using R software (version 4.4.1). The average OD readings (mean ± standard deviation, SD) for replicative measurement at different time points were calculated.

#### 4.4.2. Identification of the Subinhibitory Concentration of Amphotericin B

The dose–response curve for amphotericin B was generated to determine the substance’s subinhibitory concentration against *C. neoformans* H99. Optical density (OD) was measured at 600 nm using a spectrophotometer (Eppendorf BioSpectrophotometer Kinetic, Hamburg, Germany) after 72 h of incubation, and the subinhibitory concentration was defined as the lowest concentration that exhibited partial inhibition without fully suppressing fungal growth [30]. Data analysis and curve fitting were performed using R software (version 4.4.1).

#### 4.4.3. Screening of Naphthoquinones That Enhance the Antifungal Activity of Amphotericin B

To identify naphthoquinones showing potential synergy with subinhibitory concentrations of amphotericin B (here: 0.03 μg/mL), a preliminary screening was performed as described by Rossi et al. [30] with minor modifications. Specifically, naphthoquinones were diluted to concentrations below their minimum inhibitory concentration (MIC) values, ensuring they would not inhibit fungal growth by themselves. The finally chosen concentrations comprised lapachol and atovaquone at 200 μg/mL; 2-ClFNQ at 2 μg/mL; 2-MNQ at 1 μg/mL; and 2,3-DBNQ at 0.09 μg/mL. The *C. neoformans* H99 strain was cultured in RPMI-1640 medium with MOPS buffer at 2.5 × 10^3^ CFU/mL for these experiments. Growth was evaluated in the presence of amphotericin B (0.03 μg/mL) combined with the naphthoquinones. The experiments were conducted in triplicate, with spectrophotometric readings at 600 nm using a spectrophotometer (Eppendorf BioSpectrophotometer Kinetic, Hamburg, Germany) every 24 h for a total of 96 h.

In parallel, tubes containing only amphotericin B (0.03 μg/mL and 0.5 μg/mL) were used as controls, and tubes without any antifungal agent assessed the natural fungal growth. Additional tubes with only naphthoquinones at the above-mentioned concentrations confirmed their activity without combined application. Positive results were defined as combinations that reduced fungal growth by more than 80% [30]. The combination of 18 µg/mL erythromycin and 0.03 µg/mL amphotericin B was also tested, based on previous studies [42]. Positive and negative controls ensured the reliability of the results. Data were analyzed using R software (version 4.4.1).

#### 4.4.4. Checkerboard Assay

The interaction of amphotericin B and 2-MNQ, which showed the strongest synergistic effect in the screening, was further evaluated using the checkerboard dilution method [43,44]. The concentration range for amphotericin B was from 8 to 0.03 µg/mL. For 2-MNQ, it ranged from 8 to 0.12 µg/mL. Briefly, 50 µL of each naphthoquinone dilution (4× concentrated) was added horizontally to the wells, and 50 µL of each amphotericin B dilution (4× concentrated) was added vertically, creating a range of combinations for the tested compounds. Each well was inoculated with 100 µL of *C. neoformans* H99 inoculum (2× concentrated), resulting in a final concentration of 2.5 × 10^3^ CFU/mL. The plates were incubated at 35 °C for 72 h in a humidified incubator without additional CO_2_. Absorbance was measured at 630 nm using an iMark™ microplate reader (Bio-Rad Laboratories, Hercules, CA, USA). The fractional inhibitory concentration index (FICI) was calculated based on the following formula:$$\frac{MIC\;of\;2-MNQ\;combined}{MIC\;of\;2-MNQ\;alone}+\frac{MIC\;of\;Amb\;combined}{MIC\;of\;Amb\;alone}=FIC\;index$$

Synergy was determined if the FICI was ≤0.5, indifference was noted with FICI values between >0.5 and 4, and antagonism was identified if the FICI values were >4 as described elsewhere [27,43,45]. Data analysis was performed using R software (version 4.4.1).

#### 4.4.5. Cytotoxicity Assay

The selected naphthoquinones were subjected to cytotoxicity bioassays conducted at the Laboratory of Biological Activities (BIOPHAR) of the Faculty of Pharmaceutical Sciences—FCF at the Federal University of Amazonas (UFAM). The assays followed the protocol established by Ansar Ahmed et al. [46] and aimed at assessing cell viability in MRC-5 fibroblast cells after a 24 h exposure to the substances of interest. MRC-5 cells were seeded in 96-well plates at a density of 0.5 × 10^4^ cells per well. After a 24 h incubation period ensuring cell adhesion, the cells were treated with various concentrations of naphthoquinones, ranging from 1.56 μM to 100 μM. Subsequently, 10 μL of alamarBlue^®^ Cell Viability Reagent (Thermo Fisher Scientific, Waltham, MA, USA) was added, using a 0.4% stock solution, which was diluted 1:20 in culture medium. After a 3 h incubation period allowing resazurin metabolization, fluorescence was measured using a Bio-Rad iMark™ microplate reader (Bio-Rad Laboratories, Hercules, CA, USA) in order to quantify cell viability.

#### 4.4.6. Statistical Analysis

The results were reported as mean values ± standard deviation (SD) from three independent experiments, each conducted in triplicate. Statistical differences (*p* < 0.05) in cytotoxicity tests were determined using one-way analysis of variance (ANOVA), followed by Tukey’s or Bonferroni’s post-tests for multiple comparisons. For combination tests, Student’s *t*-test (unpaired, two-tailed) was applied to assess statistical significance. Further, Software R, version 4.4.1 (14 June 2024, using Universal C Runtime [ucrt]; https://www.R-project.org/, last accessed on 23 December 2024) was applied for data analysis and figure design.

## 5. Conclusions

This study highlights the potential of synthetic naphthoquinones, particularly 2-MNQ and 2,3-DBNQ, as potent antifungal agents against *Cryptococcus* spp. The demonstrated synergy with amphotericin B represents a significant step towards designing combination therapies, offering enhanced efficacy. These findings have important implications for future research aimed at addressing antifungal resistance and improving treatment outcomes in immunocompromised patients. Future research should focus on elucidating the molecular interactions underlying antifungal activity, investigating their mechanisms of action, conducting comprehensive preclinical evaluations, and exploring the broader therapeutic potential and applicability of these compounds across various fungal pathogens and clinical settings.

## Figures and Tables

**Figure 1 antibiotics-14-00602-f001:**
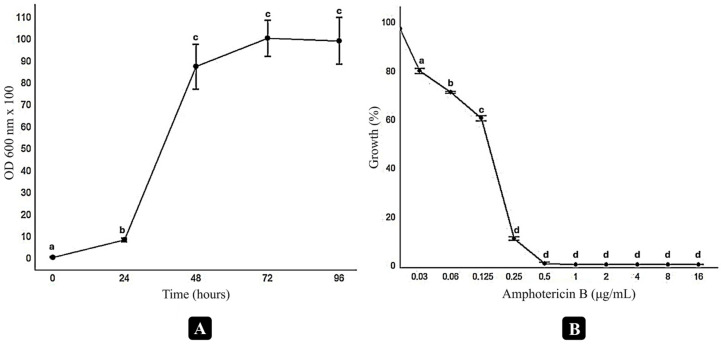
Growth and dose–response curves of *Cryptococcus neoformans* H99 under different conditions. The growth curve (**A**) was generated in RPMI medium buffered with MOPS, using an initial inoculum of 2.5 × 10^3^ CFU/mL, incubated at 35 °C with agitation at 100 rpm. Optical density (OD600 × 100) was measured at 0, 24, 48, 72, and 96 h. The dose–response curve (**B**) evaluates the inhibitory effect of varying concentrations of amphotericin B on fungal growth after 72 h of incubation. Significant differences between treatments were assessed through *t*-tests and are indicated by distinct letters above the data points (*p* < 0.05). Data analysis was performed using R software (version 4.4.1).

**Figure 2 antibiotics-14-00602-f002:**
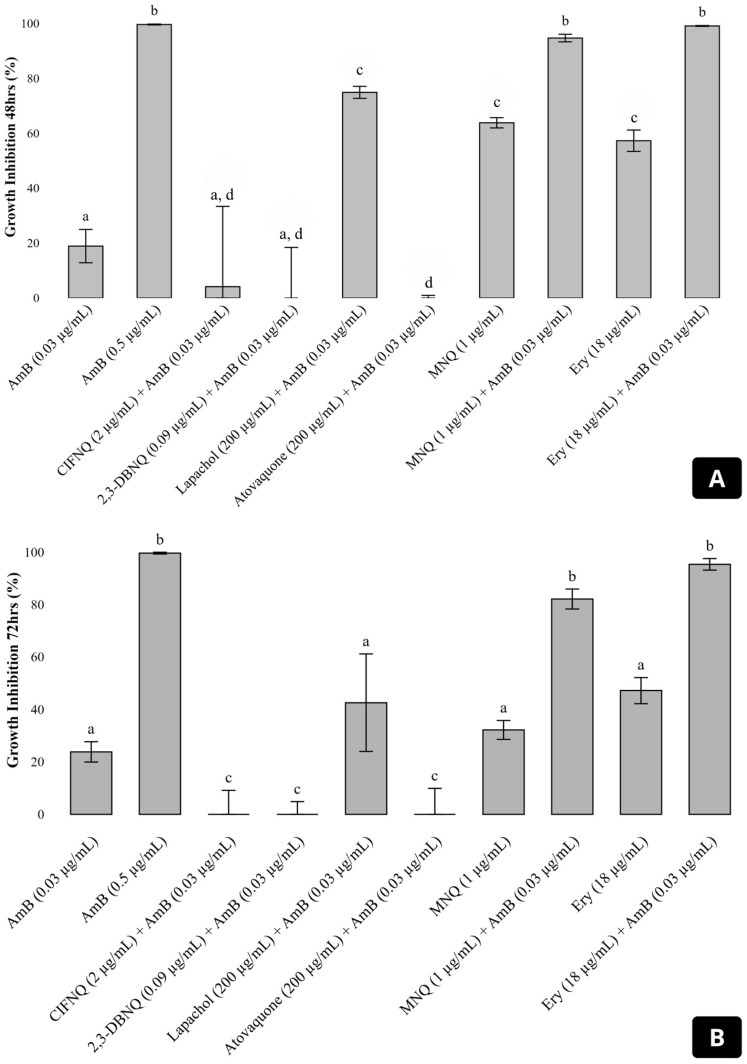
Growth inhibition of *Cryptococcus neoformans* H99 after 48 h (**A**) and 72 h (**B**) of incubation with amphotericin B (AmB) at 0.03 μg/mL, alone or in combination with various naphthoquinones and erythromycin. The graph shows the percentage of fungal growth inhibition for treatments including AmB alone, or in combination with 2-ClFNQ (2 μg/mL), 2,3-DBNQ (0.09 μg/mL), lapachol (200 μg/mL), atovaquone (200 μg/mL), 2-MNQ (1 μg/mL), and erythromycin (18 μg/mL). Error bars represent the standard deviation of the mean. Significant differences between treatments (*p* < 0.05) are indicated by different letters above the bars; treatments sharing the same letter are not significantly different. Abbreviations: 2-ClFNQ, 2-chloro-3-(2-fluoroanilino) naphthoquinone; 2,3-DBNQ, 2,3-dibromonaphthalene-1,4-dione; 2-MNQ, 2-methoxynaphthalene-1,4-dione; Ery, erythromycin.

**Figure 3 antibiotics-14-00602-f003:**
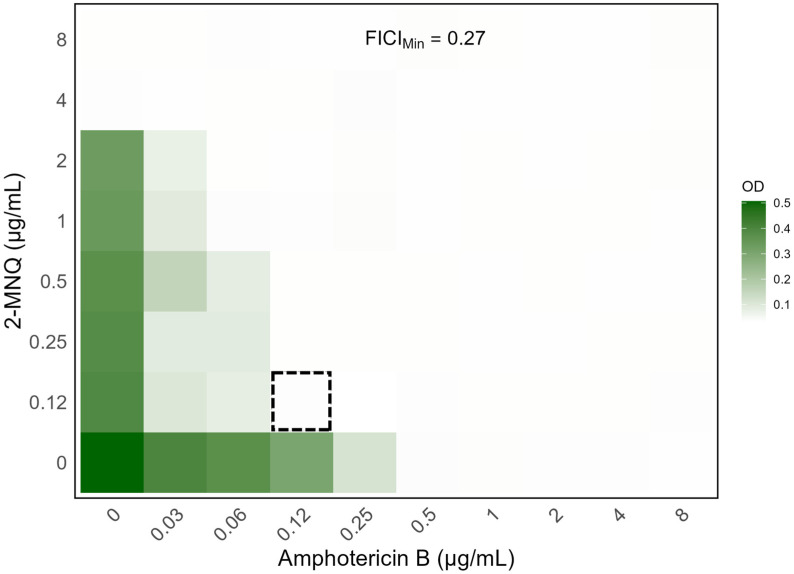
Heatmap depiction of the outputs of the checkerboard assays between 2-MNQ and amphotericin B against the *C. neoformans* H99 strain. Data represent the optical density at 630 nm of each well compared to the control for one representative biological replicate after 72 h of incubation. Gradients in the columns represent amphotericin B concentrations, while gradients in the rows represent 2-MNQ concentrations. Dark green regions represent higher cell density, while lighter regions indicate inhibition. The dashed square indicates the combination that yielded the lowest FIC index value (FICIMin = 0.27). Data analysis and figure generation were performed using R software, version 4.4.1 (https://www.R-project.org/, accessed on 23 December 2024).

**Figure 4 antibiotics-14-00602-f004:**
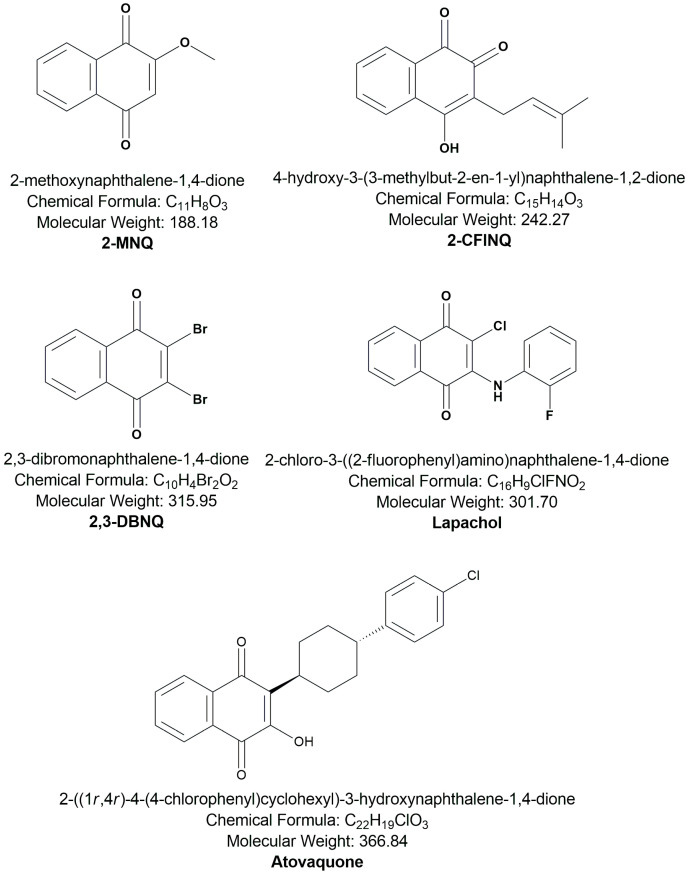
Naphthoquinones derived from 1,4-naphthoquinone investigated in the present study.

**Table 1 antibiotics-14-00602-t001:** In vitro antifungal activity of assessed naphthoquinones and amphotericin B against *C. gattii* and *C. neoformans*.

Microorganisms	Minimum Inhibitory Concentration—MIC100 (µg/mL) ^1^					
	Lapachol	2,3-DBNQ	2-MNQ	2-ClFN	ATVQ	AmB
*C. neoformans* H99	1.000	0.19	6.25	100	400	0.5
*C. neoformans* (VNI)	1.000	0.19	3.12	100	400	0.5
*C. neoformans* (VNII)	1.000	0.19	6.25	200	400	0.5
*C. neoformans* (VNIII)	1.000	0.19	12.5	100	400	0.5
*C. neoformans* (VNIV)	800	0.19	6.25	100	400	0.5
*C. gatti* (VGI)	800	0.19	6.25	100	400	0.5
*C. gatti* (VGII)	800	0.19	12.5	100	400	0.5
*C. gatti* (VGIII)	800	0.19	6.25	200	400	0.5
*C. gatti* (VGIV)	800	0.19	6.25	100	400	0.5

^1^ MIC represents the concentration, in μg/mL, of the naphthoquinone capable of inhibiting 100% of the in vitro growth of the respective isolate tested after incubation for 72 h. Lapachol: 4-hydroxy-3-(3-methylbut-2-enyl) naphthalene-1,2-dione; 2,3-DBNQ: 2,3-dibromonaphthalene-1,4-dione; 2-MNQ: 2-methoxynaphthalene-1,4-dione; 2-ClFNQ: 2-chloro-3-(2-fluoroanilino)naphthalene-1,4-dione; ATVQ: atovaquone; AmB: amphotericin B.

## Data Availability

The original contributions presented in this study are included in the article. Further inquiries can be directed to the corresponding authors.

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
