# Peer review of "Antifungal Activity of Selected Naphthoquinones and Their Synergistic Combination with Amphotericin B Against Cryptococcus neoformans H99"

_antibiotics, 2025, doi:10.3390/antibiotics14060602_

Round 1
Reviewer 1 Report
Comments and Suggestions for Authors
The study is original and contributes novel data to the field of antifungal pharmacology. Notably, there are few existing studies that explore the antifungal potential of naphthoquinones particularly in the context of cryptococcosis treatment. The manuscript relevant data on novel antifungal combinations (naphtoquinones and amphetericin), with potential implications for improving therapy in resource-limited settings or drug-resistant infections. Please kindly find below some comments/suggestions for improvement:
- While the title refers to the Cryptococcus neoformans/gattii complex, the experimental emphasis is predominantly on C. neoformans, particularly strain H99, with limited data and discussion on C. gattii isolates. I'd encourage the authors to either balance the focus across both species complexes or adjust the title to more accurately reflect the actual scope of the study.
- Cytotoxicity data: While IC50 data are given, the selectivity index (SI) (ratio of IC50 to MIC) is not discussed. This would help assess therapeutic potential. Also, please expand on whether 2-MNQ concentrations used in synergy assays fall below cytotoxic thresholds.
- A summary (preferably visual) regarding key mechanisms of action (membrane disrution, ROS generation, etc) would be helpful and enhance readers' engagement.
- Please define study limitations and potential applications in medical field with regard to cryptococcosis treatment.
Thank you.
Author Response
Comments and Suggestions for Authors
The study is original and contributes novel data to the field of antifungal pharmacology. Notably, there are few existing studies that explore the antifungal potential of naphthoquinones particularly in the context of cryptococcosis treatment. The manuscript relevant data on novel antifungal combinations (naphtoquinones and amphetericin), with potential implications for improving therapy in resource-limited settings or drug-resistant infections.
General Comment Response: Thank you for your positive and encouraging assessment of our work. We are grateful that you found the study original and relevant to the field of antifungal pharmacology. We also appreciate your recognition of the potential impact of our findings for improving treatment strategies against cryptococcosis, particularly in settings affected by antifungal resistance or limited access to current therapies.
Reviewer 1, first comment:
Please kindly find below some comments/suggestions for improvement:
While the title refers to the Cryptococcus neoformans/gattii complex, the experimental emphasis is predominantly on C. neoformans, particularly strain H99, with limited data and discussion on C. gattii isolates. I'd encourage the authors to either balance the focus across both species complexes or adjust the title to more accurately reflect the actual scope of the study.
Authors: Regarding your first specific suggestion, we agree that the experimental focus of the study was primarily on Cryptococcus neoformans, especially strain H99. Therefore, we have revised the title to better reflect this scope. The updated title is: “Antifungal activity of selected naphthoquinones and their synergistic combination with amphotericin B against Cryptococcus neoformans H99.”
This change can be found on page 1, line 1-3 of the revised manuscript.
Reviewer 1, second comment:
Cytotoxicity data: While IC50 data are given, the selectivity index (SI) (ratio of IC50 to MIC) is not discussed. This would help assess therapeutic potential. Also, please expand on whether 2-MNQ concentrations used in synergy assays fall below cytotoxic thresholds.
Authors: We agree with this important suggestion. In response, we have included a discussion on the selectivity index (SI) for the compound 2-MNQ in the revised manuscript. The SI was calculated as the ratio between the ICâ‚…â‚€ value in MRC-5 human fibroblast cells (2.25 µg/mL) and the MIC values against Cryptococcus neoformans (ranging from 3.12 to 12.5 µg/mL), resulting in SI values between 0.18 and 0.72. Although these values are below the commonly accepted threshold of SI ≥ 10 for high selectivity (Cos et al., 2006; Leocádio et al., 2025), we clarified that the concentration of 2-MNQ used in the synergy assays (1 µg/mL) was below the ICâ‚…â‚€, indicating that antifungal activity was achieved without exceeding in vitro cytotoxic levels. Furthermore, despite the low SI, structural optimization of 2-MNQ could further improve its selectivity profile. This addition can be found on page 7, lines 210 to 217 of the revised manuscript.
- Cos P, Vlietinck AJ, Berghe DV, Maes L. Anti-infective potential of natural products: How to develop a stronger in vitro ‘proof-of-concept’. J Ethnopharmacol. 2006;106(3):290–302.
https://doi.org/10.1016/j.jep.2006.04.003
- Leocádio VAT, Miranda IL, Magalhães MHC, Dos Santos Júnior VS, Goncalves JE, Oliveira RB, Maltarollo VG, Bastos RW, Goldman G, Johann S, Teixeira de Aguiar Peres N, Santos DA. Thiazole Derivatives as Promising Candidates for Cryptococcosis Therapy. ACS Infect Dis. 2025 Mar 14;11(3):639-652.
https://doi.org/10.1021/acsinfecdis.4c00732
Reviewer 1, third comment:
A summary (preferably visual) regarding key mechanisms of action (membrane disrution, ROS generation, etc) would be helpful and enhance readers' engagement.
Authors: Thank you for your suggestion. We recognize the relevance of illustrating mechanisms of action to enhance comprehension. However, the mechanistic aspects mentioned in our manuscript—such as membrane disruption and ROS generation—are based on previously published findings (e.g., Almeida et al., 2023) and were not directly investigated in our experiments. We therefore opted not to further elaborate on these aspects visually, to maintain alignment with the scope of our study. This clarification has been incorporated into the limitations section to clearly define the boundaries of the scope of our experimental approach. This addition can be found on page 7, final paragraph, lines 229–239 of the revised manuscript.
Reviewer 1, fourth comment:
Please define study limitations and potential applications in medical field with regard to cryptococcosis treatment.
Authors: Agreed. We have, accordingly, revised the manuscript to better define the main limitations of the study and to contextualize the potential applications of our findings regardng the treatment of cryptococcosis. Although the antifungal activity and synergistic potential of 2-MNQ were confirmed in vitro, the molecular mechanisms underlying this synergy were not explored, and the toxicity was assessed using only one human cell line. Moreover, mechanistic aspects such as membrane disruption and ROS generation were based on prior reports and not experimentally investigated in this study. These limitations have now been clearly addressed in the final paragraph of the discussion section. This change can be found on page 7, final paragraph, lines 229–239 of the revised manuscript.
Reviewer 2 Report
Comments and Suggestions for Authors
This review offers a well-rounded and timely overview of recent advances in the use of bacterial pigments for textile dyeing, with a valuable focus on their antibacterial properties. The manuscript is clearly structured, draws on a wide range of relevant literature, and addresses a compelling interdisciplinary topic that brings together microbiology, materials science, and sustainability. That said, I do have a few questions for the authors and suggestions that I believe could help strengthen the paper further.
Questions:
- How do bacterial pigments compare in durability and colourfastness with synthetic dyes under industrial wash and light conditions?
- Is there any cost-benefit analysis between bacterial pigment extraction and synthetic dye production?
- What are the existing or anticipated regulatory barriers for industrial-scale implementation of bacterial pigments in medical textiles?
- Are there any documented cytotoxicity or allergenic risks associated with prolonged exposure to fabrics dyed with these pigments?
- Can current extraction methods (especially green techniques) be effectively scaled for commercial textile production?
Suggestions for improvement
- The manuscript references a wide range of studies, many of which provide numerical data such as pigment yields, dye exhaustion rates, and colourfastness scores. Compiling this information into a comparative summary table could greatly enhance readability and offer a clearer overview of the performance metrics across different pigments and methods.
- While the sustainability potential of bacterial pigments is well discussed, the review could benefit from a brief comparison, whether qualitative or quantitative, between these bio-based dyes and conventional synthetic dyes. This might include life-cycle assessments, production costs, or environmental trade-offs to help contextualize the advantages of bacterial pigments.
- If the authors included examples of pilot-scale trials or early industrial applications would significantly strengthen the practical relevance of the review.
- The manuscript discusses various biosynthetic pathways and extraction methods in detail. However, these descriptions would be even more accessible if supported by clearly labelled and prominently placed diagrams or schematics. Visual aids would help readers better understand complex mechanisms and processes described in the text.
Minor errors
Line 190 and line 331: “trough” should be “through.”
Author Response
General Comment Response: We would like to respectfully inform the editorial team that the comments and suggestions provided by Reviewer 2 appear to refer to a different manuscript, specifically one related to bacterial pigments and their use in textile dyeing. Therefore, we believe these comments were erroneously forwarded to us. We respectfully ask the editor to cross check on our respective impression and to let us know on how to proceed regarding this likely misunderstanding.
Reviewer 3 Report
Comments and Suggestions for Authors
In this short communication, the authors probed into a possibility that the naphthoquinones, especially synthetic ones, could function as effective antifungal agents with synergistic properties when combined with amphotericin B (AmB). With the model fungal infection selected as Cryptococcosis (induced by Cryptococcus spp.), the synergy effect was demonstrated with a checkerboard heatmap. A proposed mode of action was also provided: while AmB disrupting fungal cell membranes through ergosterol binding, 2-MNQ would intensify this effect by increasing membrane permeability and inducing oxidative stress.
With the criteria for publication likely met, the authors were encouraged to consider the following perspectives for improve the brief argument's understanding to readers:
1) essentially, more evidence on the synergistic effect - what appeared now on the heatmap would be arguably rather, additive effect;
2) SEM, or TEM evidence on the fungal pyroptosis/necroptosis process upon single/combinatorial treatment to support the proposed mode of action differences between 2-MNQ and AmB, further elucidating the possibility of synergistic effect.
Also a typo, on line 35: "Six selected naphthoquinones..." should be Five.
Author Response
Comments and Suggestions for Authors
In this short communication, the authors probed into a possibility that the naphthoquinones, especially synthetic ones, could function as effective antifungal agents with synergistic properties when combined with amphotericin B (AmB). With the model fungal infection selected as Cryptococcosis (induced by Cryptococcus spp.), the synergy effect was demonstrated with a checkerboard heatmap. A proposed mode of action was also provided: while AmB disrupting fungal cell membranes through ergosterol binding, 2-MNQ would intensify this effect by increasing membrane permeability and inducing oxidative stress. With the criteria for publication likely met, the authors were encouraged to consider the following perspectives for improve the brief argument's understanding to readers.
General Comment Response: We sincerely thank the reviewer for the thoughtful and constructive comments. We appreciate your positive evaluation of our study and your suggestions aiming at improving the clarity and scientific foundation of our work. Below, we address each point individually and provide clarification or justification as applicable.
Reviewer 3, first comment: Essentially, more evidence on the synergistic effect - what appeared now on the heatmap would be arguably rather, additive effect.
Authors: Thank you for this observation. We understand that, based on the visual representation alone (heatmap), the interaction might appear additive. However, as clarified by Odds (2003) and further emphasized in pharmacological models such as Loewe’s additivity and Bliss independence, the additive effect represents the baseline from which synergy is detected (Chou, 2006). In our study, the combination of 2-MNQ and amphotericin B against Cryptococcus neoformans H99 yielded a FICI value of 0.27, which is in agreement with the synergy threshold (≤0.5), speaking in favor of a true synergistic interaction rather than simple additivity. Although the heatmap may not visually reflect this effect, the quantitative data support a synergistic antifungal effect. We have clarified this point in the revised discussion section, page 7, lines 187–197 of the revised manuscript.
Chou TC. Theoretical basis, experimental design, and computerized simulation of synergism and antagonism in drug combination studies. Pharmacol Rev. 2006 Sep;58(3):621-81. doi: 10.1124/pr.58.3.10.
Reviewer 3, second comment: SEM, or TEM evidence on the fungal pyroptosis/necroptosis process upon single/combinatorial treatment to support the proposed mode of action differences between 2-MNQ and AmB, further elucidating the possibility of synergistic effect.
Authors: We acknowledge the reviewer's valuable suggestion regarding the use of ultrastructural analyses such as scanning electron microscopy (SEM) or transmission electron microscopy (TEM) for the investigation of cell death mechanisms (e.g., pyroptosis or necroptosis) in Cryptococcus neoformans upon treatment with 2-MNQ, amphotericin B, or their combination. These advanced imaging techniques can indeed provide further mechanistic insights into the morphological alterations associated with antifungal activity and synergy.
Unfortunately, however, funding constraints did not allow respective experiments for the here presented study. Our author team hopes to be able to perform them in a future follow-up study, if sufficient funding can be provided. However, this limitation is now explicitly addressed in the revised discussion section of the manuscript (page 7, final paragraph, lines 229–239). We hope to present detailed ultrastructural findings in the course of a future project, which should specifically focus on mode-of-action assessments of the proposed interaction.
Reviewer 3, third comment: Also a typo, on line 35: "Six selected naphthoquinones..." should be Five.
Authors: Thank you for pointing out this typographical error. We have corrected the sentence on line 35 to accurately reflect that five naphthoquinones were evaluated in the study.
Round 2
Reviewer 1 Report
Comments and Suggestions for Authors
The revised manuscript has taken into account the previously-given suggestions.
Author Response
General Comment Response: We appreciate the reviewer’s feedback and are pleased to know that the revisions addressed the previously given suggestions.
Reviewer 2 Report
Comments and Suggestions for Authors
This study evaluates the antifungal properties of five naphthoquinone derivatives against Cryptococcus neoformans and Cryptococcus gattii, identifying 2-methoxynaphthalene-1,4-dione (2-MNQ) as the most promising candidate, with MICs ranging from 3.12 to 12.5 µg/mL. Notably, 2-MNQ demonstrated synergistic effects when combined with amphotericin B (AmB), reducing the MIC of AmB by over fourfold (FICI = 0.27). Although the selectivity index, based on MRC-5 fibroblast cytotoxicity data, was below the standard safety threshold, the concentrations used in combination assays were non-cytotoxic. The authors propose 2-MNQ as a potential adjuvant antifungal agent, while emphasizing the need for further mechanistic and in vivo validation. That said, I have several questions and suggestions that could help strengthen the manuscript.
- Could the authors provide mechanistic validation (e.g., ROS measurement, membrane disruption assays) to support the hypothesized mechanisms behind the synergistic effect between 2-MNQ and AmB?
- Given the low selectivity index (SI < 1), what is the rationale for considering 2-MNQ a promising compound? Could additional cell lines or primary cells be tested to strengthen cytotoxicity claims?
- How do these in vitro findings translate into clinical relevance? Are there any in vivo models or preclinical studies planned?
- The study mentions genetic diversity among tested Cryptococcus Could the authors elaborate on how representative these are of clinical infections globally?
- Was the stability of 2-MNQ during the 72-hour assay confirmed? The reduction in antifungal activity over time suggests possible degradation or metabolic alteration.
Minor suggestions:
- Line 32: Correct "assed" to "assessed".
- Line 275: The phrase "remains unclear and warrants further investigation" would benefit from specifying potential future experiments.
- The word "promising" appears often—consider varying the language with more precise terms like “synergistic”, “potent”, or “effective”.
Author Response
Comments and Suggestions for Authors
This study evaluates the antifungal properties of five naphthoquinone derivatives against Cryptococcus neoformans and Cryptococcus gattii, identifying 2-methoxynaphthalene-1,4-dione (2-MNQ) as the most promising candidate, with MICs ranging from 3.12 to 12.5 µg/mL. Notably, 2-MNQ demonstrated synergistic effects when combined with amphotericin B (AmB), reducing the MIC of AmB by over fourfold (FICI = 0.27). Although the selectivity index, based on MRC-5 fibroblast cytotoxicity data, was below the standard safety threshold, the concentrations used in combination assays were non-cytotoxic. The authors propose 2-MNQ as a potential adjuvant antifungal agent, while emphasizing the need for further mechanistic and in vivo validation. That said, I have several questions and suggestions that could help strengthen the manuscript.
General Comment Response: We sincerely thank the reviewer for the thoughtful and constructive feedback. We appreciate your detailed evaluation and helpful suggestions, which contributed to improving the scientific clarity and consistency of our manuscript. Please find our point-by-point responses below.
Reviewer 2, first comment: Could the authors provide mechanistic validation (e.g., ROS measurement, membrane disruption assays) to support the hypothesized mechanisms behind the synergistic effect between 2-MNQ and AmB?
Authors: We thank the reviewer for the valuable observation. Indeed, the present manuscript focuses specifically on the screening for antifungal activity and the synergistic interaction between 2-MNQ and amphotericin B. The mechanistic hypothesis discussed—based on previous findings published by our group (Almeida JDR, Fonseca RSK, Sousa NSO, Cortez ACA, Lima ES, Oliveira JGS, Souza ÉS, Frickmann H, Souza JVB. Antifungal potential, mechanism of action, and toxicity of 1,4-naphthoquinone derivatives. Eur J Microbiol Immunol. 2024, 14(3), 289–95. https://doi.org/10.1556/1886.2024.00072)—is not derived from experiments conducted in this study. This clarification has been included in the final part of the Discussion: We have acknowledged this as a limitation in the revised Discussion section (page 8, lines 276–280), and we are actively working to secure funding for future studies that will include these mechanistic assessments.
Reviewer 2, second comment: Given the low selectivity index (SI < 1), what is the rationale for considering 2-MNQ a promising compound? Could additional cell lines or primary cells be tested to strengthen cytotoxicity claims?
Authors: We agree with this important observation. Although the selectivity index (SI) values obtained for 2-MNQ are below the threshold generally accepted for high selectivity (SI ≥ 10), we emphasize that the concentration used in the synergy assays (1 µg/mL) was consistently below the ICâ‚…â‚€ value determined in MRC-5 fibroblasts, indicating that antifungal activity was achieved without exceeding cytotoxic levels in vitro. Moreover, the simple and functional chemical structure of 2-MNQ favors future structural modifications that could improve its safety margin, reinforcing its potential as a promising antifungal adjuvant in combination therapies. To further support its therapeutic applicability, we recognize the importance of expanding cytotoxicity assessments by including additional cell lines and primary human cells. These experiments are planned as part of our follow-up work and will be conducted as soon as appropriate funding becomes available. This limitation has been incorporated into the revised manuscript (page 8, lines 246–255 and 280–282).
Reviewer 2, third comment: How do these in vitro findings translate into clinical relevance? Are there any in vivo models or preclinical studies planned?
Authors: We appreciate the reviewer’s concern regarding the translational relevance of our findings. This is a critical aspect in antifungal drug development, and we agree that bridging the gap between in vitro activity and clinical applicability requires a systematic and well-supported approach. The in vitro synergy observed between 2-MNQ and amphotericin B nevertheless suggests therapeutic potential against cryptococcosis. However, before progressing to in vivo studies, we acknowledge a need to further expand our in vitro investigations, including cytotoxicity assessments using additional cell lines and primary human cells, as well as mechanistic studies to clarify the compound’s mode of action. These steps are essential to ensure a safe and evidence-based transition to animal testing. In vivo studies using a murine model of cryptococcal infection are planned as a later stage of the project, depending on the availability of financial resources to support more complex experimental designs. These points are now clearly addressed in the revised Discussion section (page 8, lines 280–284).
Reviewer 2, fourth comment: The study mentions genetic diversity among tested Cryptococcus Could the authors elaborate on how representative these are of clinical infections globally?
Authors: We thank the reviewer for this important question. Our panel consisted of eight genetically distinct reference strains — four C. neoformans strains (VNI, VNII, VNIII, VNIV) and four C. gattii strains (VGI, VGII, VGIII, VGIV) — in addition to the well-characterized clinical reference strain C. neoformans H99. These molecular types correspond to the genotypes most frequently reported worldwide and are recognized by the WHO and several large-scale epidemiological studies as the dominant lineages recovered from clinical, environmental, and veterinary sources. Notably, these genotypes exhibit marked differences in virulence, geographic distribution, and antifungal susceptibility, which reinforces the importance of comparative studies on them. This information has been clarified in the revised Methods section (page 9, lines 309–313), with the inclusion of the appropriate bibliographic references.
Reviewer 2, fifth comment: Was the stability of 2-MNQ during the 72-hour assay confirmed? The reduction in antifungal activity over time suggests possible degradation or metabolic alteration.
Authors: We thank the reviewer for this insightful question. The chemical stability of 2-MNQ during the 72-hour assay was not directly assessed in this study. Indeed, we observed a marked reduction in the antifungal activity of 2-MNQ when used alone, with 63.84% inhibition at 48 hours and only 32.20% at 72 hours. This decline may be related to compound degradation, fungal adaptation, or metabolic inactivation. In contrast, the combination of 2-MNQ (1 µg/mL) and amphotericin B (0.03 µg/mL) maintained high inhibitory activity over time (94.72% at 48 h and 82.14% at 72 h), clearly differing from the drop observed for 2-MNQ alone. Further, amphotericin B alone at the same concentration showed only ~20% inhibition at both time points. These considerations were acknowledged, and the absence of compound stability assessment was explicitly indicated as a study limitation in the Discussion section (pages 7 and 8, lines 192–201 and 285-287, respectively).
Minor suggestions:
Reviewer: Line 32: Correct "assed" to "assessed".
Response: Thank you for noting this typographical error. The word “assed” has been corrected to “assessed” in line 32 of the revised manuscript.
Reviewer: Line 275: The phrase "remains unclear and warrants further investigation" would benefit from specifying potential future experiments.
Response: Thank you for this valuable suggestion. To address the reviewer’s comment, we revised the text to specify the experimental approaches that could clarify the mechanisms underlying the observed synergy. Future studies should incorporate ROS quantification, membrane integrity assays and ultrastructural analysis using electron microscopy, such as scanning electron microscopy (SEM) or transmission electron microscopy (TEM), and SYTO/PI (SYTO Green and Propidium Iodide) staining to visualize morphological features of fungal cell death (e.g., pyroptosis or necroptosis). These strategies would help to validate the proposed mode of action. This revision has been included in the Discussion section (page 8, lines 276–280) of the revised manuscript.
Reviewer: The word "promising" appears often—consider varying the language with more precise terms like “synergistic”, “potent”, or “effective”.
Response: We appreciate this helpful observation. We reviewed the manuscript and replaced redundant occurrences of the word “promising” with more precise terms such as “synergistic,” “potent,” and “effective,” depending on the context. Revisions were made in the Introduction (line 68), the Results (line 130), and the Discussion (lines 213 and 269) sections.